# Parity and Interval from Previous Delivery—Influence on Perinatal Outcome in Advanced Maternal Age Parturients

**DOI:** 10.3390/jcm10030460

**Published:** 2021-01-26

**Authors:** Amir Naeh, Mordechai Hallak, Rinat Gabbay-Benziv

**Affiliations:** 1Obstetrics and Gynecology Department, Hillel Yaffe Medical Center, Hadera 38100, Israel; mottih@hy.health.gov.il (M.H.); gabbayrinat@gmail.com (R.G.-B.); 2The Rappaport Faculty of Medicine, Technion, Haifa 32000, Israel

**Keywords:** advanced maternal age, adverse pregnancy outcome, elderly gravida, interpregnancy interval, nulliparity, pregnancy complications

## Abstract

Objective: To investigate the effect of parity and interpregnancy interval (IPI) on perinatal outcomes in advanced maternal age (AMA) parturients. Methods: A population-based retrospective cohort study of all women older than 40 years, who had a singleton live birth after 24 weeks in the United States in 2017 Women were categorized to three groups by parity and interval from last delivery: primiparas, multiparas with IPI ≤ 5 years, and multiparas with IPI > 5 years. Primary outcome was composite adverse neonatal outcome (preterm delivery <34 weeks, birthweight <2000 g, neonatal seizure, neonatal intensive care unit admission, Apgar score <7 at 5 min, or assisted ventilation >6 h). Secondary outcome was composite adverse maternal outcome and other adverse perinatal outcomes. Univariate and multivariate analysis were used to compare between groups. Results: During 2017, 3,864,754 deliveries were recorded into the database. Following exclusion, 109,564 AMA gravidas entered analysis. Of them, 24,769 (22.6%) were nulliparas, 39,933 (36.4%) were multiparas with IPI ≤ 5 years, and 44,862 (40.9%) were multiparas with IPI > 5 years. Composite neonatal outcome was higher in nulliparas and in multiparas with IPI > 5 years, in comparison to multiparas with IPI ≤ 5 years (16% vs. 13% vs. 10%, respectively, *p* < 0.05). Maternal composite outcome was similar between groups. In the multivariable analysis, relative to nulliparas, only multiparity with IPI ≤ 5 years had a protective effect against the composite neonatal outcome (aOR 0.97, 95% CI 0.95–0.99, *p* < 0.001). Conclusion: Among AMA gravidas, multiparity with IPI ≤ 5 years has a significant protective effect against adverse neonatal outcomes when compared to nulliparas. Multiparity with IPI > 5 years is no longer protective.

## 1. Introduction

During the last few decades, there has been an increasing trend for child bearing in the later reproductive years, particularly in high-income countries [1]. As age is a continuum rather than a categorical variable, the definition of advanced maternal age is not solid, and most studies refer to women over 35 to 40 years as such. Advanced maternal age gravidas have higher rates of adverse pregnancy outcomes including: pre-eclampsia, gestational diabetes mellitus, preterm birth, fetal growth restriction, cesarean delivery, stillbirth, and more [2,3,4,5,6].

Other than age, nulliparity is also associated with adverse pregnancy outcome [7,8], therefore, multiparity (assuming normal outcome) is considered protective. In multipara women, data suggests that interpregnancy interval (IPI), whether short (below 18 months) or long (above 60 month) has an impact on both maternal and neonatal outcome [9,10,11,12,13,14,15]. It is thought that following prolonged IPI, the maternal, physiologic, and anatomical pregnancy adaptations gradually decline and become comparable to those at their first pregnancy [9].

Therefore, the aim of this study was to investigate the combined effect of parity and IPI on perinatal outcomes in advanced maternal age parturients.

## 2. Materials and Methods

We performed a population-based cohort study of all parturients older than 40 years at time of delivery with singleton live birth at 2017 in the United States. We used de-identified natality data assembled by the National Vital Statistics System of the National Center for Health Statistics that provides demographic and health data for births occurring during the calendar year in the United States (available at: https://data.nber.org/data/vital-statistics-natality-data.html) [16]. The U.S. Standard Certificate of Live Birth, issued by the U.S. Department of Health and Human Services, has served for many years as the principal means for attaining uniformity in the content of the documents used to collect information on births in the United States. In 2003, birth certificates were revised to contains more detailed demographic, medical, and obstetric data compared to the previous 1989 version. The revised birth certificate was gradually adopted by the states and from 2016, represents 100% live births in the 50 states and Washington, DC.

Only livebirth deliveries at 24 or more gestational weeks with neonatal weights above 500 g were included. We excluded women if maternal age less than 40 years at time of delivery, carrying multifetal gestation, or had any known fetal anomalies or chromosomal abnormalities. Additionally, all births with unknown data on the time interval from previous delivery were excluded. This study was exempt from review by the institutional review board at our institution because the data we used do not meet the criteria for human subject research by federal standards. The STROBE (Strengthening the Reporting of Observational Studies in Epidemiology) guidelines for reporting observational studies were followed [17].

All parturients that entered analysis were categorized to three groups. Group 1 included nulliparas parturients. Group 2 included all multiparas with previous live birth delivery documented within 5 years from current delivery. Group 3 included multiparas with previous live birth delivery occurring at longer than 5 years interval from current delivery.

The primary outcome that was evaluated between groups was a composite of adverse neonatal outcome that included: preterm delivery <34 weeks, birthweight <2000 g, neonatal seizure, neonatal intensive care unit admission (NICU), Apgar score less than 7 at 5 min, or neonatal assisted ventilation for more than 6 h. Secondary outcomes included composite adverse maternal outcome (including uterine rupture, unplanned hysterectomy, maternal intensive care unit admission, and maternal blood transfusion) and other adverse perinatal outcomes including gestational diabetes, hypertensive disorders during pregnancy, induction of labor, mode of delivery, gestational age at delivery and birthweight, and any one of the variables included in the composite primary outcome separately. Newborns or women with more than one adverse outcome were counted once when formulating the composites.

### Statistical Analysis

Statistical analysis was performed using R software (R version 3.6.2, R Foundation for Statistical Computing, Vienna, Austria). For univariate analyses of baseline differences, Student *t*-test and chi-square test were used for continuous data and categorical data, respectively. Multivariable logistic regression was performed to adjust the composite outcomes to potential confounders: maternal age, pregestational diabetes, chronic hypertension, smoking, race, body mass index, use of assisted reproductive technology, gestational diabetes, gestational hypertension, preeclampsia, and cesarean delivery. *p*-value was considered statistically significant of <0.05.

## 3. Results

During 2017, 3,864,754 live births were recorded into the database. In 124,574 (3.2%) of them, maternal age was older than 40 years at the time of delivery. Following exclusion, 109,564 cases met the inclusion criteria and entered analysis (Figure 1).

Almost quarter of the women delivering after 40 years of age were nulliparas (24,769/109,564, 22.6%). The rest were multiparas; 39,933 (36.4%) had their last previous delivery within 5 years from the current pregnancy and 44,862 (40.9%) had their last delivery at more than 5 years interval.

Maternal characteristics stratified by study group are presented in Table 1. For the entire cohort median maternal age was 41 years. For multipara women, median number of previous deliveries was four (range: 2–8). Group 2 had a mean IPI of 34 months and group 3 had mean IPI of 130 months. Maternal characteristics differed in race, smoking, BMI, presence of chronic hypertension or diabetes, and obstetrical history (previous preterm or cesarean deliveries). Surprisingly, there was no difference in assisted reproductive use between nulliparas and multiparas.

Perinatal outcomes of the three study groups are shown in Table 2. Overall, for the entire cohort, 12.4% presented at least one adverse neonatal composite outcome (13,621/109,564). In the univariate analysis, neonatal composite outcome was statistically different between groups, i.e., nulliparas women had the highest rate of composite neonatal adverse outcome (16%), followed by multiparas with IPI longer than 60 months (13%).

Multiparas women with previous delivery at less than 5 years interval had the lowest rate of composite adverse neonatal outcome (10%, *p* < 0.05). For individual neonatal adverse outcomes, neonates of nulliparas had highest rates of low Apgar scores (<7 at 5 min), NICU admission, and required more assisted ventilation.

Only 0.4% (427/109,564) gravidas presented at least one of the composite adverse maternal outcomes with similar distribution between study groups. Nulliparas (Group 1) had highest rates of gestational hypertension (10.8% vs. 5.8% vs. 8.3%) and preeclampsia (0.6% vs. 0.2% vs. 0.4%) during pregnancy, with higher rates of induction of labor (35.2% vs. 22.1% vs. 26.1%) and cesarean deliveries (57.9% vs. 41.8% vs. 44.6%). Postpartum, they received more blood transfusions compared to other groups (0.6% vs. 0.4% vs. 0.5%), Group 1, 2, and 3 respectively.

Overall, Group 2 delivered later in pregnancy (38.3 vs. 38.5 vs. 38.1, gestational weeks) the largest babies (3169 vs. 3379 vs. 3324 g) Group 1, 2, and 3, respectively, *p* < 0.05 for all. For the majority of adverse outcomes, Group 3 was second to Group 1, leaving lowest rates of complications among multiparas with previous delivery within 5 years interval.

Results of multivariable analysis are shown in Table 3. Previous delivery within 5 years difference had a significant protective effect against the composite neonatal outcome (aOR 0.97, 95% confidence interval 0.95–0.99, *p* < 0.001) relative to nulliparas. Previous delivery at longer than 5 years was no longer protective from composite neonatal outcome.

## 4. Discussion

In this study, we aimed to evaluate the combined effect of parity and IPI on perinatal outcomes in advanced maternal age parturients. Our main findings were (1.) composite adverse neonatal outcome among advanced maternal age gravidas was overall high (12.4% of the cohort). Highest rate of adverse neonatal composite outcome was seen in nulliparas, followed by multiparas with IPI longer than 5 years and lastly, in multiparas with IPI within 5 years (16% vs. 13% vs. 10%, *p* < 0.05). Separate neonatal outcomes (low Apgar scores, NICU admission, assisted ventilation) were also more common among nulliparas. (2.) Nulliparas had higher rates of separate adverse maternal outcomes including hypertensive disorders of pregnancy, induction of labor, cesarean deliveries, and postpartum blood transfusions. However, parity had no influence on composite adverse maternal outcome. (3.) Utilizing a multivariable analysis and relative to nulliparas, a previous delivery within 5 years interval had a significant protective effect against the composite neonatal outcome. This effect did not persist when delivery interval was longer than 5 years.

Advanced maternal age is a well-established risk factor for adverse pregnancy outcomes [2,3,4,5,6]. Nulliparity adds further risk [7,8], putting the advanced maternal age nullipara gravida at the focus of research studies and pregnancy surveillance. Consistent with this assumption, multiparity is considered a protective factor among all maternal ages, including among advanced maternal age parturients. In this study, we aimed to evaluate this assumption stratified by the time interval from last delivery.

Reports differentiating older primiparas from multiparas are scarce. Shechter M.G. et al. [8] conducted a retrospective study that aimed to evaluate the impact of parity on adverse perinatal outcome among advanced maternal age parturients (over 35 years at time of delivery). The authors demonstrated higher rates of multifetal pregnancies, preterm deliveries, hypertensive disorders, diabetes, and fetal growth restriction among nulliparas compared to multiparas gravidas. Although their study demonstrated the protective effect of parity on risk for pregnancy complications, they did not take into account the time interval from last delivery for the multiparas women. Our study, not only refined the effect of age, including only parturients over 40 years, but also evaluated the protective effect of parity with regard to the time interval elapsed from last delivery.

Several studies have demonstrated an association between long IPI and pregnancy complications, including pre-eclampsia and eclampsia, preterm delivery, small for gestational age neonates, birth defects, higher rates of cesarean deliveries, and lower success rate for trial of labor after cesarean [9,10,11,12,13,14,18,19,20,21,22]. A recent study has also found that long IPI is associated with increased risk for long-term neurological morbidity of the offspring [15]. To note, there is no universal definition for long IPI, although most studies use the cutoff of 5 years. In our study, parity had a protective effect only when last delivery occurred within 5 years from the index pregnancy.

Various explanations were offered for the association between long IPI and pregnancy complications, including an age-related, physiological mechanisms, and other causes. Zhu et al. found that the optimal IPI for preventing adverse perinatal outcomes is 18–23 months [9]. They offered two hypotheses for the association between long IPI and pregnancy complications. The first was that vascular, physiological, and anatomical changes in pregnancy help parturients to gain growth-supporting capacities. If another fetus is not conceived for a long period of time, those capacities may gradually decline, and thus causing maternal physiologic characteristics to become more similar to those of the nullipara gravida. Second, with advanced time, metabolic or anatomical factors may cause both delayed fertility and adverse birth outcomes. Secondary infertility by itself is associated with an increased incidence of preterm birth [23]. Another possible explanation is that pregnancies conceived after a long IPI have higher probability of being unplanned, and they are more common in women with a low socioeconomic status, a significant risk factor for adverse pregnancy outcomes [24,25].

Our results demonstrate that IPI of less than 5 years had a significant protective effect against adverse neonatal outcomes. Since our study cohort included only women delivering after 40 years of age, this effect cannot be attributed to the parturients age. In addition, rates of assisted reproductive use were similar between the groups, excluding it as the cause for our results. BMI and rates of smoking, chronic hypertension, and pregestational diabetes were higher in the group with IPI > 5 years, and although adjusted in the multivariate analysis, they might also have contributed to the higher adverse outcomes in this group. The finding that this protective effect was lost in the group with IPI longer than 5 years is more compatible with a physiological mechanism, rather than an association with a categorical factor. We suggest that maternal physiological adaption to pregnancy, mainly vascular (e.g., increased uterine blood flow, reduced systemic resistance, and elevated cardiac output) is a fundamental process that is not everlasting and decreases over time. If this process is not regenerated, with time, the plasticity of these components declines, and therefore the protective effect is lost.

Our study also demonstrated the effect of parity and IPI on maternal adverse outcome. Similar to the neonatal outcomes, advanced maternal age nulliparas, as compared to multiparas, were at higher risk for hypertensive disorders of pregnancy, induction of labor, cesarean deliveries, and postpartum blood transfusions. Interestingly, multiparas with IPI > 5 years had significantly higher rate of gestational hypertension in comparison to multiparas with IPI < 5 years (8.3 vs. 5.3, *p* < 0.001). However, the overall impact of parity as well as of IPI for the mother, seems less significant compared to the impact on adverse neonatal outcome. One possible explanation for the diminished effect may be the association of maternal outcomes with age itself, unlike neonatal or pregnancy complications that may be more related to the maternal adaptation to pregnancy, other than to the absolute age-related risk [26].

Our study has several strengths. It is a population-based study that includes more than 100,000 parturients older than 40 years. The large sample size of our study enables evaluation of risk in subgroups within an overall high-risk cohort of parturients. In addition, the diverse population that derives from using a national data-base, with the use of multivariable logistic regression analysis allows us to control for risk factors that are already known to be associated with each adverse outcome. However, our study has also limitations, majority of them attributed to its retrospective design. The study is representative of all live birth in the United States in 2017 and, therefore, includes medical centers with heterogenicity in practice patterns that may have an effect on at least some of the measured outcomes. In addition, we excluded multiple gestations and pregnancies complicated by intrauterine fetal death, which is no doubt an important adverse outcome ignored in this study. Data on assisted reproductive therapy were available to only small part of the cohort and were possibly influenced by reporting bias. Lastly, this study is also prone to limitations of vital statistics data, which include likely underreporting of maternal comorbidities or other adverse outcomes.

## 5. Conclusions

In summary, our study demonstrates that among advanced maternal age gravidas, a previous delivery within 5 years interval has a significant protective effect against adverse neonatal outcomes when compared to nulliparas. In case of IPI of more than 5 years, this effect is lost, and the risk for adverse neonatal outcomes becomes comparable to that of advanced maternal age nullipara. We suggest that an IPI longer than 5 years should be considered as a significant risk factor for pregnancy complications and needs to be combined with other maternal characteristics when evaluating the individualized risk for adverse pregnancy complications. Early identification of parturient with an increased risk will enable to facilitate targeted surveillance and early intervention. Specifically, advanced maternal age gravidas with IPI > 5 years should be considered high-risk for pregnancy complications. Ideally, they should undergo prepregnancy consultation for possible lifestyle modifications to improve outcomes, and during pregnancy, they should be addressed with higher surveillance, including prophylactic aspirin treatment, close follow-up to detect fetal growth restriction, and serial monitoring for preterm labor and preeclampsia.

## Figures and Tables

**Figure 1 jcm-10-00460-f001:**
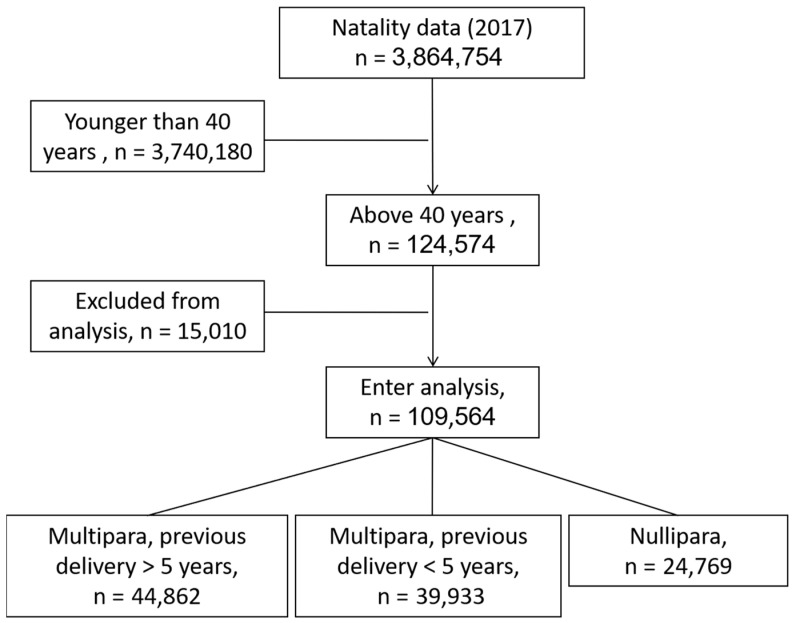
Flow chart of patients in each cohort after applying exclusion and inclusion criteria.

**Table 1 jcm-10-00460-t001:** Maternal characteristics stratified by parity and interval from previous delivery.

Maternal Characteristic	Group 1: Nulliparas *n* = 24,769	Group 2: Multiparas, Previous Delivery <5 Years *n* = 39,933	Group 3: Multiparas, Previous Delivery <5 Years *n* = 44,862	*p*-Value
Maternal age, years	41.65 ± 1.98	41.34 ± 1.67	41.54 ± 1.74	**<0.001**
Number of previous deliveries	-	4.46 ± 2.08 Median 4 (range: 2–8)	4.14 ± 1.72 Median 4 (range: 2–8)	**<0.001**
Previous delivery interval, months	-	34.16 ± 12.53	130 ± 57.12	**<0.001**
Race:				
White	17,661 (71%)	29,661 (74%)	31,042 (69%)	
Black	3193 (13%)	5315 (13%)	8152 (18%)	
American Indian or Alaskan Native	99 (0%)	384 (0%)	268 (0.7%)	
Asian or Pacific Islander	3816 (16%)	5284 (12%)	4689 (12%)	**<0.001**
Assisted reproduction *	3515 (82%)	1819 (82%)	1064 (80%)	0.14
Smoking	554 (2.2%)	1018 (2.6%)	2229 (5%)	**<0.001**
BMI (kg/m^2^)	26.62 ± 6.45	27.11 ± 6.32	28.10 ± 6.53	**<0.001**
Pregestational diabetes	511 (2.1%)	648 (1.6%)	1172 (2.6%)	**<0.001**
Chronic hypertension	1217 (4.9%)	1361 (3.4%)	2533 (5.6%)	**<0.001**
Previous preterm delivery	-	2548 (6.4%)	2509 (5.6%)	**<0.001**
Number of previous cesarean deliveries	-	1.48 ± 0.87	1.53 ± 0.78	**<0.001**

Continuous variables are presented as mean ± SD or median (range); categorical values are *n* (%). Statistically significant *p*-values as marked in bold. * Data on ART was available for only part of the cohort; percentages are calculated from valid data.

**Table 2 jcm-10-00460-t002:** Perinatal outcome stratified by parity and interval from previous delivery.

Maternal Characteristic	Group 1: Nulliparas *n* = 24,769	Group 2: Multiparas, Previous Delivery <5 Years *n* = 39,933	Group 3: Multiparas, Previous Delivery <5 Years *n* = 44,862	*p*-Value
Pregnancy complications:				
Gestational diabetes	3108 (12.6)	4630 (11.6)	6790 (15.1)	**<0.0001**
Gestational hypertension	2675 (10.8)	1301 (5.8)	3707 (8.3)	**<0.0001**
Preeclampsia	143 (0.6)	98 (0.2)	173 (0.4)	**<0.0001**
Induction of labor	8712 (35.2)	8828(22.1)	11,695 (26.1)	**<0.0001**
Cesarean delivery	14,344 (57.9)	16,269 (41.8)	19,994 (44.6)	**<0.0001**
Maternal blood transfusion	147 (0.6)	165 (0.4)	207 (0.5)	**0.004**
Uterine rupture	5 (0)	20 (0.1)	20 (0)	0.11
Maternal ICU admission	81 (0.3)	85 (0.2)	132 (0.3)	**0.01**
Unintended hysterectomy	27 (0.1)	72 (0.2)	60 (0.1)	0.05
Gestational age at delivery	38.3 ± 2.43	38.5 ± 2.07	38.1 ± 2.33	**<0.0001**
Birthweight	3169 ± 603	3379 ± 562	3324 ± 597	**<0.0001**
Birthweight by category:				
<1500 g	451 (1.8)	288 (0.7)	641 (1.4)	
1500–2499 g	2284 (9.2)	1935 (4.8)	3433 (7.7)	
>2500 g	22,034 (89)	37,710 (94.4)	40,788 (90.9)	**<0.0001**
Male gender	12,545 (50.6)	20,300 (50.8)	22,841 (50.9)	0.79
Apgar score:				
0–3	132 (0.5)	146 (0.4)	223 (0.5)	
4–6	472 (1.9)	426 (1.1)	641 (1.4)	
7–8	3433 (13.9)	4314 (10.9)	5249 (11.7)	
9–10	20,658 (83.7)	34,847 (87.7)	38,604 (86.3)	**<0.0001**
Neonatal ICU admission	3320 (13.4)	3218 (8.1)	4807 (10.7)	**<0.0001**
Neonatal seizures	8 (0)	9 (0)	14 (0)	0.69
Neonatal assisted ventilation >6 h	472 (1.9)	500 (1.3)	691 (1.5)	**<0.0001**
Neonatal composite	3869 (16)	3884 (10)	5868 (13)	**<0.001**
Maternal composite	104 (0.4)	145 (0.4)	178 (0.4)	0.5

Continuous variables are presented as mean ± SD; categorical values are *n* (%). Statistically significant *p*-values as marked in bold. Composite adverse neonatal outcome includes: preterm delivery <34 weeks, birthweight < 2000 g, neonatal seizure, neonatal intensive care unit admission, Apgar score less than 7 at 5 min or neonatal assisted ventilation for more than 6 h. Composite adverse maternal outcome includes uterine rupture, unplanned hysterectomy, maternal intensive care unit admission, and maternal blood transfusion.

**Table 3 jcm-10-00460-t003:** Adverse composite neonatal outcome evaluated by multivariable analysis to adjust for confounders.

	aOR B(EXP)	95% Confidence Interval	*p*-Value
Maternal group			
Group 1	Reference		
Group 2	0.97	0.95–0.99	**<0.001**
Group 3	0.99	0.97–1.02	0.651
Maternal age	1	0.99–1	0.154
Race:			
White	Reference		
Black	1.02	0.99–1.06	0.153
American Indian or Alaskan Native	0.94	0.79–1.11	0.471
Asian or Pacific Islander	0.99	0.97–1.02	0.804
Pregestational diabetes	1.16	1.08–1.24	**<0.001**
Chronic hypertension	1.13	1.08–1.17	**<0.001**
Smoking	1.27	1.1–1.46	**0.001**
BMI, kg/m^2^			
<18.5	0.99	0.94–1.05	0.830
18.5–24.9	Reference		
25–29.9	1.02	1–1.04	**0.045**
30–34.9	1.03	1–1.06	**0.022**
35–39.9	1.03	0.99–1.06	0.151
>40	1.02	0.97–1.07	0.476
Assisted reproductive	1.01	0.99–1.03	0.524
Gestational diabetes	1.02	0.99–1.04	0.129
Gestational hypertension	1.12	1.09–1.15	**<0.001**
Preeclampsia	1.25	1.11–1.40	**<0.001**
Cesarean delivery	1.05	1.03–1.06	**<0.001**

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
