# Peer review of "Parity and Interval from Previous Delivery—Influence on Perinatal Outcome in Advanced Maternal Age Parturients"

_jcm, 2021, doi:10.3390/jcm10030460_

Round 1
Reviewer 1 Report
The authors investigated maternal and neonatal outcome of women who gave birth at advanced age.
I think this is an interesting and well designed paper, it is written very well and the statistical analysis is well done.
However, I have a few comments:
- According to table 1, all women of group 3 seem to have several risk factors for adverse obstetric outcome, such as smoking, chronic hypertension and a higher BMI compared to women of group 1 and 2. This was mentioned in one sentence in "results" only. I think that the adverse obstetric outcome in group 3 might be associated with the maternal characteristics in table 1.
- I believe that the "take home message" is a little poor. What can we learn from these results ? All these pregnancies , and especially women from group 1 and 3 are "high risk pregnancies" and should be treated as such. Women at advanced age with the wish to get pregnant, should be seen before pregnancy for "life style modifications" and "prepare for pregnancy". Additionally they should be seen on a regular basis throughout pregnancy and should be screened for preelampsia, fetal growth, preterm delivery etc. I believe that the authors should focus a little more on an optimized management in such high risk pregnancies
Reviewer 2 Report
This is a well written manuscript and although it does not provide any novel scientific information, it has a lot of information, excellent statistical analysis, clear results and therefore merits publication, although there are a few points, which need to be addressed by the authors:
- The collected data are from year 2017. Why such delay to submit the manuscript? Was not possible to retrieve data for more recent period?
- It is strange that the percentage of ART was similar among the three groups. Since more women undergo ART at more advanced reproductive age and usually they remain at one child, what is your explanation on this?
- The hypertension was almost double (5.8 vs 8.3). Please make a comment
Round 2
Reviewer 1 Report
Thank you for the changed manuscript- well done !
Author Response
Thank you